# Analysis of a Municipal Solid Waste Disposal Site: Use of Geographic Information Technology Tools for Decision Making

**Juan Antonio Araiza-Aguilar** [1,*]**, María Neftalí Rojas-Valencia** [2]**, Hugo Alejandro Nájera-Aguilar** [1]**, Rubén Fernando Gutiérrez-Hernández** [3] **and Carlos Manuel García-Lara** [1]

1   School of Environmental Engineering, University of Science and Arts of Chiapas, North Beltway 1150, Lajas Maciel, Tuxtla Gutiérrez 29000, Mexico; hugo.najera@unicach.mx (H.A.N.-A.); carlos.garcia@unicach.mx (C.M.G.-L.)

2   Institute of Engineering, National Autonomous University of Mexico, External Circuit, University City, Mexico City 04510, Mexico; nrov@pumas.iingen.unam.mx

3   Chemical and Biochemical Engineering Department, National Technology of Mexico—Technological Institute of Tapachula, Tapachula 30700, Mexico; rub.gutierrez@tapachula.tecnm.mx

*   Correspondence: juan.araiza@unicach.mx; Tel.: +52-961-617-0440

**Abstract:** In this study, the operation of a final disposal site for municipal solid waste in the state of Chiapas, in Mexico, was evaluated. Several spatial analyses and Geographic Information Technology (GIT) tools were used. It was found that the site's current operation and location are deficient, partially complying with regulations. The gaseous dispersion is not far-reaching (from 100 to 8725 $\mu g/m^3$ for landfill gas, and from 0.01 to 0.35 $\mu g/m^3$ for $H_2S$) but requires attention to avoid olfactory unpleasantness. Liquid emissions (conservative pollutants) move in the east direction of the final disposal site, which can damage the environmental infrastructure (water supply wells) in the long term. The highest and lowest concentrations were found in years 1 (12,270 $mg/m^3$) and 20 (1080 $mg/m^3$), respectively. Thermal emissions around the dumping site are important due to the formation of microclimatic zones. Temperature differences were found during the analysis period, ranging from 8.37 °C in summer to 2.49 °C in winter, which are due to waste decomposition processes and anthropogenic activities. Finally, the change in land use around the dumping site increased at a rate of 5.82% per year, mainly due to the growth of homes, communication routes, and shopping centers.

**Keywords:** municipal solid waste; spatial analysis; final disposal site

## 1. Introduction

The effective management of municipal solid waste (MSW) is an important challenge in today's society. Factors such as population growth, consumption habits, and migration have a significant impact on the amount of waste produced in the world [1,2]. In developed countries, MSW management policies and hierarchies are aimed at preventing and minimizing, while in developing countries they are reversed, which means that the disposal of waste through sanitary landfills is still at a fundamental stage [3,4]. According to Espinosa et al. [5], Hereher et al. [6], and Owusu et al. [7], final disposal using sanitary landfills is very popular in many countries, especially in Latin America and the Caribbean, mainly due to the low construction and operating costs compared to other waste disposal technologies.

Sanitary landfills that are properly operated must have comprehensive control systems, which include leachate collection facilities, landfill-gas venting systems, surface and groundwater monitoring stations, specialized equipment, and machinery, etc. [8,9]. However, many times, the location, construction, and operational procedures are deficient, which causes this final disposal technology to become an open dump, which severely affects the quality of the environment and the health of the human being.

Several studies have reported significant damage in the surroundings or within a final disposal site (FDS). In Gouveia and Prado [10], Jiang et al. [11], and Palmiotto et al. [12], the affectations of the human being are addressed, which range from olfactory discomfort to various types of cancer. In Vongdala et al. [13], Przydatek [14], and Kumar et al. [15], affectations are analyzed for other components of the environment, for example, the soil, water, air, vegetation, and fauna. All these affectations are called risk that is associated with the management of MSW [16]. In addition, they normally occur due to liquid and gaseous emissions (leaching and landfill gas), which are generated in the decomposition processes of organic matter and by the ingress of rainwater [17].

Various international regulations specify that the FDS must be constantly monitored from their start-up to their closure or abandonment [18,19]. However, the municipal authorities in charge of operating these infrastructures do not have the economic resources to carry it out, especially in small municipalities or towns in developing countries. Furthermore, continuous monitoring schemes require suitable equipment and laboratory tests that consume time and effort [9]. For this reason, researchers have chosen to use other viable methods to monitor, evaluate, collect, and generate information on waste dump sites, for example, through the use of Geographic Information Technologies (GITs).

GITs were first developed in the 1960s. They are commonly integrated by Geographic Information Systems, remote sensing techniques, and satellite positioning systems [20]. Today, GITs are used in the different stages of MSW management. Karimi et al. [21], Lacoboaea and Petrescu [22], and Mahmood et al. [23] use them to monitor the temperature and health status of the vegetation around a dump site. Amal et al. [24], Fennonato et al. [25], and Kinobe et al. [26] use them to model efficient waste collection routes. Finally, Khorsandi et al. [27], Damasceno et al. [28], Ağacsapan and Cabuk [29], and Araiza et al. [30] use these technologies both for the location of treatment infrastructure or final disposal, and to model the dispersion of pollutants emitted and the effects produced.

The aim of this study was to evaluate the operation of an MSW FDS in Chiapas, Mexico, using various spatial analysis techniques and GIT tools. The results of this manuscript can serve decision making in environmental matters, and also as a basis for developing future work in the study area. Public databases are used, as well as specific fieldwork to collect information. Initially, the site conditions, its construction, and operational characteristics are analyzed. Subsequently, analyses are carried out to determine the spatial scope of the emission of leachate, landfill gas, and temperatures, as well as changes in land use in the surroundings of the FDS. Finally, compliance with Mexican regulations and the international literature is reviewed.

## 2. Materials and Methods

### 2.1. Study Area

The FDS analyzed is located in the state of Chiapas, southeast of Mexico, particularly in the municipality of Reforma. Its location coordinates are $93°10'7.07''$ W and $17°51'48.49''$ N, 5 km west of the municipal seat (Figure 1). The territorial extension of the study area has grown notably, influenced not only by the high rates of waste deposited, but also by the operation of the authorities in charge. Currently, the municipal government proposes to restructure the environmental policy, particularly solid waste, for which the evaluation of the location and operation of the current FDS is urgently required.

### 2.2. Field Data Collection

In order to evaluate the construction conditions of the FDS, as well as compliance with environmental regulations, the Mexican standard NOM-083-SEMARMAT-2003 [19] was used. This information was complemented by field visits to the dumping site and the consequent review of the existing infrastructure (Figure 2a). In this investigation, work was also carried out to obtain the waste composition (Figure 2b), the volumetric weights, and the waste entry rates, particularly the procedures outlined in NMX-AA-015-1985 [31], NMX-AA-019-1985 [32], NMX-AA-022-1985 [33], and Araiza et al. [34].

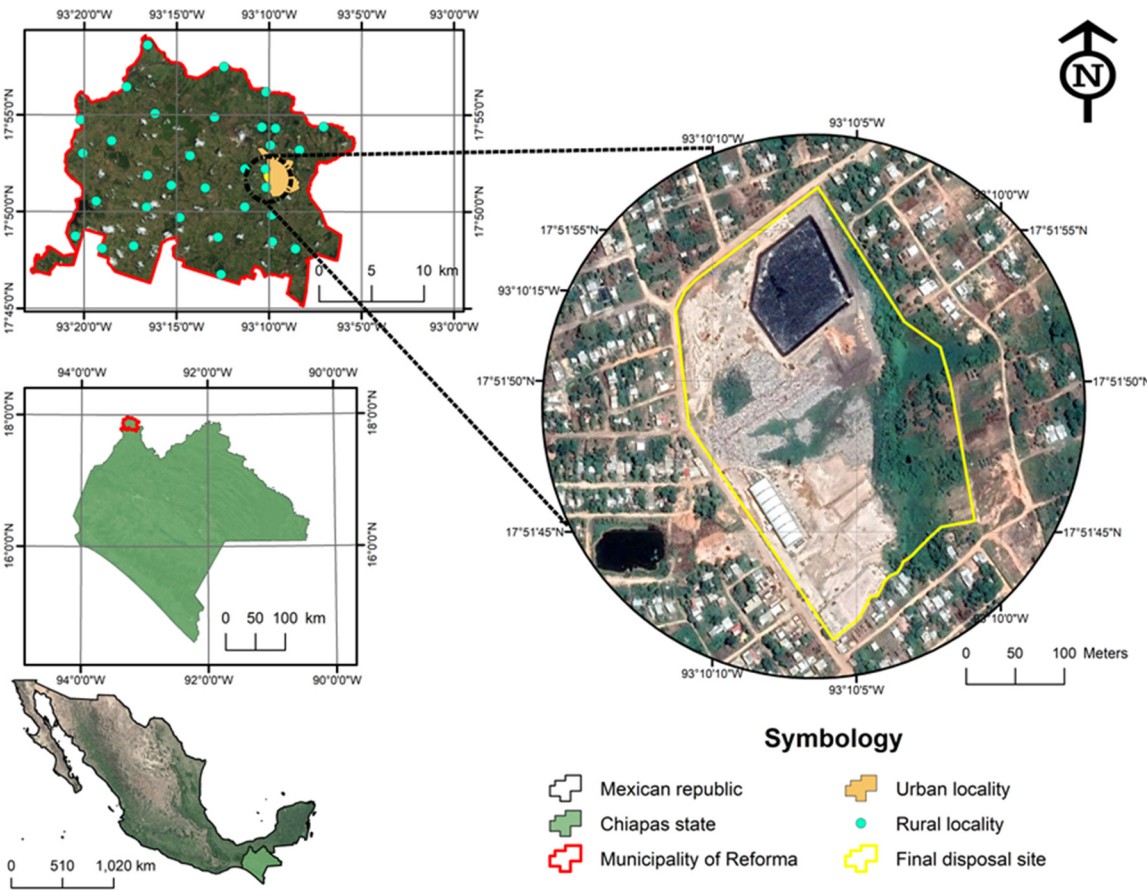

**Figure 1.** Study area: municipality of Reforma, state of Chiapas, Mexico.

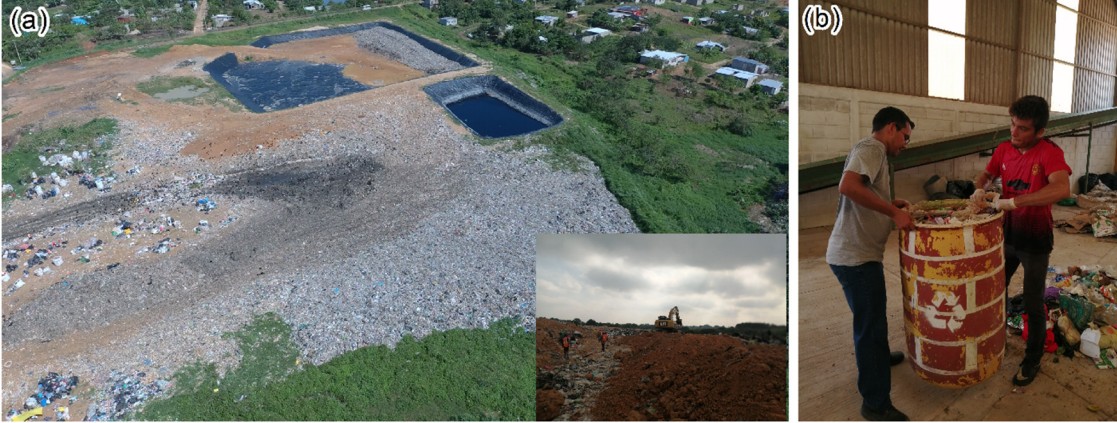

**Figure 2.** Survey information in the field: (**a**) infrastructure and operation of the dumping site; (**b**) waste parameters and characteristics.

### 2.3. Spatial Analysis and Use of GIT

In order to evaluate the possible affectations beyond the limits of the FDS, different spatial analysis techniques with the support of GIT tools were applied, particularly distance measurements and Land Use and Land Cover Change (LULC), as well as analysis of the dispersion of leachates, landfill gas, and land surface temperature (LST).

Distance measures and other restrictions refer to the distance between the FDS and the elements that may be affected, such as population settlements, streams and bodies of water,

airports, and protected natural areas, among others. All these distances and restrictions must be respected because they are established in international environmental regulations (see OJF, [19]). In this paper, the aforementioned distances were considered, as well as distances to other elements necessary for proper operation, for example, distance to roads or buyers of plastics and metals. In order to determine these distances, buffer geoprocesses were used in the GIS environment, as well as several references that have compiled this information (Table 1), specifically, the works of Araiza et al. [35], Al-Ruzouq et al. [36], and Abd-El Monsef and Smith [37].

**Table 1.** Distances and restrictions of the FDS with respect to other infrastructures.

| Variables | Distances/Restriction | Source |
|---|---|---|
| Airports | 13 km | |
| Protected natural areas | Stay out of them | |
| Population settlements | 500 m | OJF, [19] |
| Surface water | 500 m | Araiza et al. [35] |
| Water extraction wells | 500 m | Al-Ruzouq et al. [36] |
| Distance to roads | 500 m | Abd-El Monsef and |
| Buyers nearby | 1000 m | Smith [37] |
| Current land uses | Stay out of non-compatible land uses | |
| Faults and fractures | Stay out of them | |

The LULC measures are related to the degree of occupation of the land surface by some type of vegetation, but also by allocations derived from human activity [38]. In this investigation, LULC is used to determine the changes inside and outside the FDS. High spatial resolution Google Earth images from the years 2005, 2013, 2015, and 2021 were used, which had been previously georeferenced. These dates are the most representative of the behavior of the dump site. The year 2005 represents the opening year and 2013 and 2015 are transition years for the operation of the FDS, while 2021 is the year with the most recent data on operation. An Object-Based Image Analysis (OBIA) was applied, since these analyses offer greater benefits than the classic pixel-based classification methods, mainly due to the richness of the image in terms of colors, shapes, texture, and characteristics of the surroundings of the evaluated landscape [39,40]. The OBIA method began with the segmentation of images through Segment Mean Shift, and later the classification of the image using the ArcGIS Training Sample Manager with four classes (water, urban/anthropized, tree coverage, and shrub and herbaceus coverage), as well as Interactive Supervised Classification, which speeds up the maximum likelihood classification process. Finally, to estimate the rates of change between the different years and classes, the cross-tabulation matrices proposed by Pontius et al. [41], as well as the metrics of FAO [42], were used. The workflow is shown in Figure 3.

Landfill gas dispersion measurements were made to determine the spatial scope and possible concentrations of gaseous emissions in the surroundings of the FDS. In particular, the production rates of landfill gas and $H_2S$ were modeled, since they are two of the main gases generated in landfills. The dispersion analysis was carried out in two seasons (dry and rainy) through the AERMOD modeling package, particularly using its three modules (AERMET, AERMAP, and AERMOD). Meteorological data such as direction, wind speed, air temperature, atmospheric pressure, relative humidity, and cloud cover were processed by AERMET. The topographic and receiver data of the emissions were processed with AERMAP. The results of the previous processes, together with the information on the emission rates, were analyzed in AERMOD. The input data were obtained from different sources. The meteorological information (2018–2021) was provided by CONAGUA [43], especially the data from the automatic meteorological station closest to the study area. The topographic information was obtained from a digital elevation model provided by INEGI [44]. The gas emission rates generated in the FDS were determined by means of

the Mexican biogas model version 2.0, specifically through the methodology proposed by Castillo et al., [45] and Araiza and Rojas [46]. The workflow is shown in Figure 4.

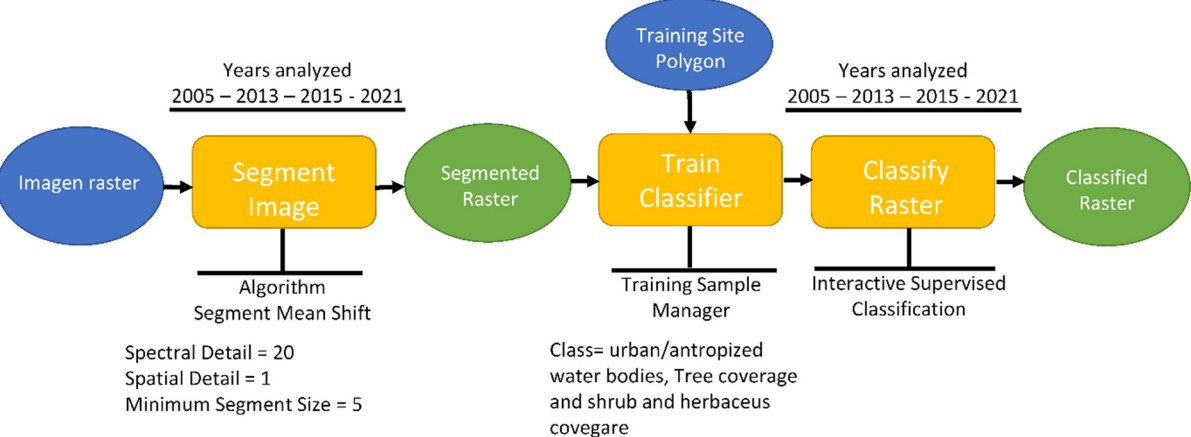

**Figure 3.** Workflow of LULC analysis.

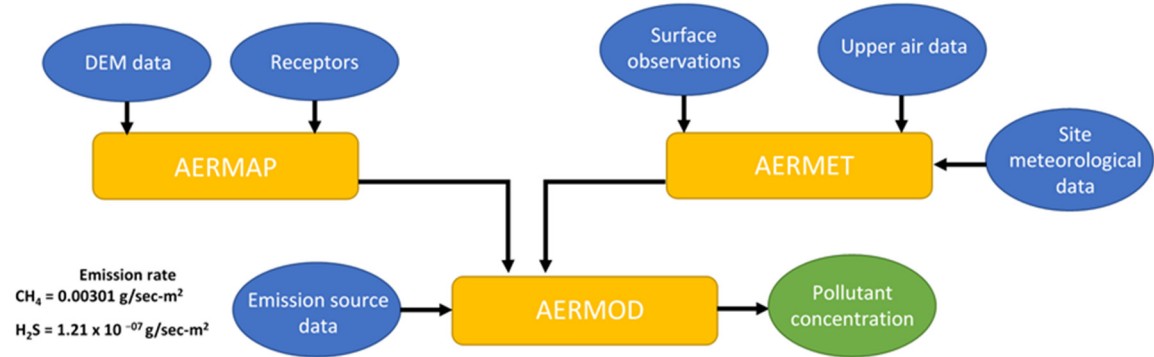

**Figure 4.** Workflow of landfill gas dispersion analysis.

The measurement of leachate dispersion is another important parameter to measure the proper functioning of an FDS, especially when it operates as an open dump or controlled site, which generally do not have the infrastructure to control such emissions. In this research, the ArcGis 10.X Groundwater Toolset is used to determine the movement and direction of groundwater flow, as well as the spatial and temporal variations in the movement of the plume of a conservative contaminant (Ion Chloride $Cl^-$). The tool used allows simplified advection–dispersion modeling of the groundwater components present in the study area [47]. Guleria et al. [48] suggest using these simplified pollutant transport modeling methods, because they require less data and use data that are readily available. The workflow initially consisted of determining groundwater seepage velocity from groundwater head, porosity, transmissivity, and aquifer thickness data using Darcy flow function. Subsequently, particle tracking is determined using groundwater flow data. Finally, the concentration distribution in the subsoil was calculated using the porous puff function. The starting data were obtained from open databases, the specialized literature, and empirical determinations. The characteristics of the aquifer of the study area and the lithological data were obtained from CONAGUA [49] and SGM [50]. The groundwater head was generated from the piezometric network data from CONAGUA [51], and also by using a deterministic interpolation method. Other simulation-specific data are shown in Figure 5.

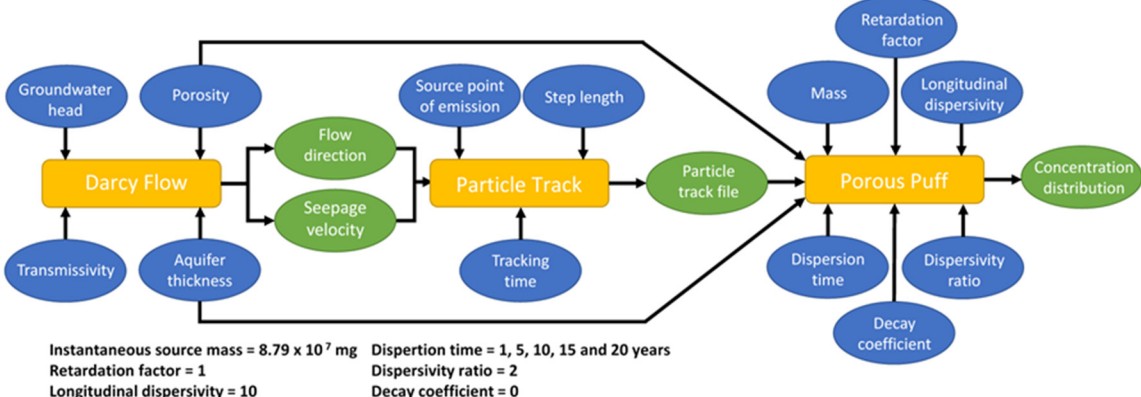

**Figure 5.** Workflow of leachate dispersion analysis.

The LST measurement is the last parameter obtained through GIT tools. LST is widely used in environmental studies because it provides useful information about the physical properties of the surface and climate, either on a regional or global scale [52,53]. The determination of LST in this paper is significant, because FDS are responsible for the formation of microclimatic zones around them, which result in many local environmental implications [54]. The determination of LST was made from the thermal bands of the Landsat-7 TM (band 6) and LANDSAT-8 OLI (Band 10) satellites, both for the month of May (summer season) and December (winter season). The analysis was carried out for the years 2005, 2013, 2015, and 2021, since those years are representative of the behavior of the dump site. The LST derivation procedures were carried out as established in the papers by Das et al. [55] and Morsy and Aboelkhair [56]. The workflow is shown in Figure 6.

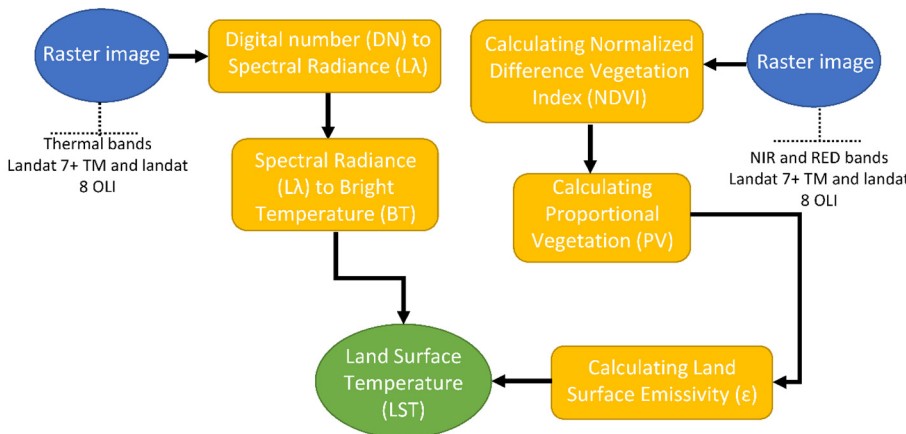

**Figure 6.** Workflow of LST analysis.

## 3. Results and Discussion

### 3.1. Operation of the FDS

The FDS of Reforma, Chiapas, has operated since 2005 as an open dump site (Figure 7a), in breach of what is indicated in the Mexican norm NOM-083-SEMARMAT-2003 [19]. Between 2005 and 2019, MSW was deposited directly onto the ground, in an approximate area of 2.00 Ha, without guaranteeing the daily coverage of the waste (Figure 7b), as well as the collection, conduction, or extraction of leachate and landfill gas. In some periods of time, these wastes were also burned, causing scenarios of air pollution and bad odors. Today, the waste still present is accommodated in a smaller area and is covered with soil material (Figure 7c); however, leachate production and migration are still notorious. Since 2019, the operation of the FDS is like a controlled site, because new cells have been installed (Figure 7d), which have leachate control systems (artificial barrier with geomembranes) that

guarantee extraction and storage for subsequent evaporation (Figure 7e). The discharge cell has a capacity of 25,000 m³ (Figure 7f) and good management, so it can have a useful life of approximately 2 years. Additionally, other basic control structures have been added, such as the perimeter fencing and the incorporation of machinery for excavation and covering waste.

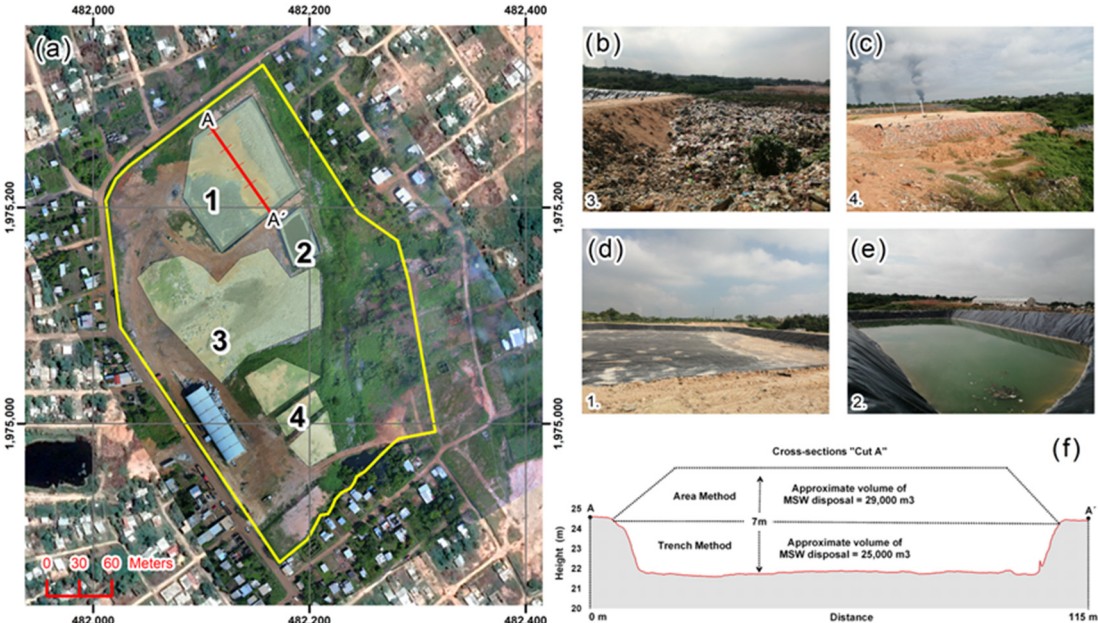

**Figure 7.** Operation of the FDS of Reforma, Chiapas: (**a**) landfill infrastructure; (**b**) waste dump area without cover; (**c**) disposal cell with final coverage; (**d**) current landfill cell; (**e**) leachate lagoon; (**f**) useful volume for waste confinement.

This dumping site receives waste not only from the 50 towns in the Reforma municipality, but also from other nearby municipalities such as Juárez and Pichucalco. On average, 47.08 t/day of MSW arrive, of which 73.02% (34.38 t/day) comes from Reforma, particularly from household sources and public services. An amount of 6.21% (2.93 t/day) also comes from Reforma, but from its commercial sources. Finally, the rest comes from various sources of waste production in the municipalities of Juárez and Pichucalco (20.77%, equivalent to 9.78 t/day). Other data such as MSW income per day can be seen in Figure 8.

On the other hand, the composition of the deposited waste was mostly organic (34.27%). There is also the presence of other materials such as metals (1.76%), plastics (16.27%), cardboard (11.53%), and glass (3.46%), but unfortunately only a small percentage of these by-products (<5%) are separated, stored, treated, or sold (Figure 9). These actions decrease the useful life of the current dump cells. Regarding the volumetric weights of disposed waste, these vary from 0.217 t/m³ for recently arrived waste to 0.60 t/m³ for compacted waste.

It is important to mention that other structures still need to be incorporated for a correct operation of the FDS, such as storm drains, road improvement, construction of cells for emergencies, natural barriers, etc. Their non-incorporation can cause physical instability in the waste dumping processes and severe affectations in neighboring towns. For example, studies by Ouyang et al. [57], Laner et al. [58], and Han et al. [59] address risk scenarios due to lack of infrastructure in the FDS, such as landslides, floods, or very severe contamination processes of surface and groundwater, mainly due to the discharge of nutrient salts, organic matter, and heavy metals.

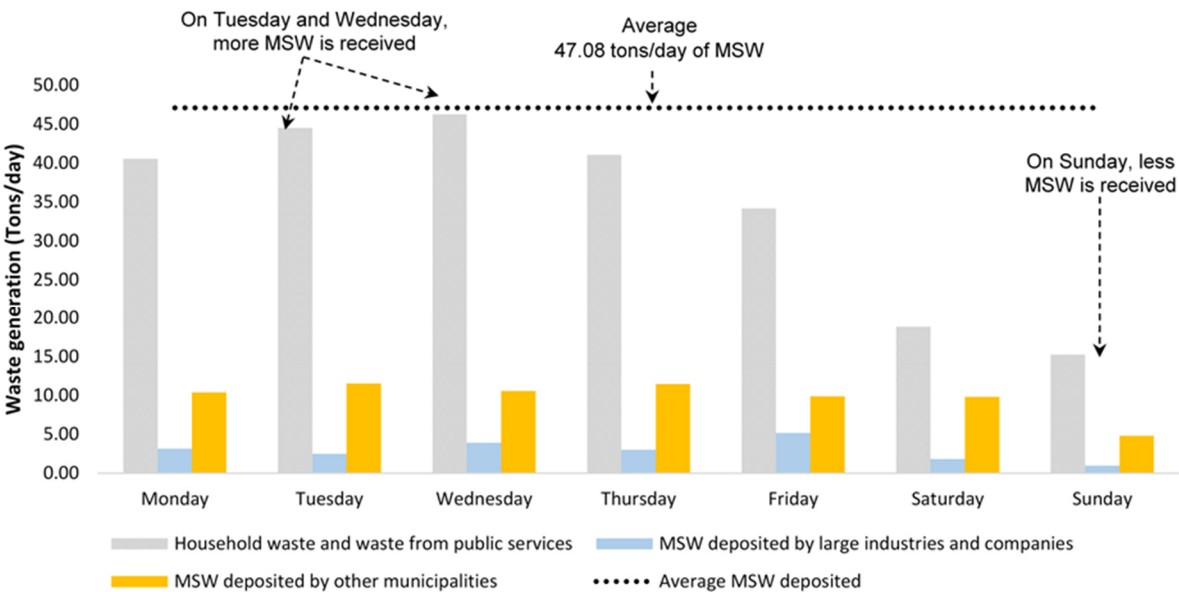

**Figure 8.** MSW deposited in the FDS of Reforma, Chiapas.

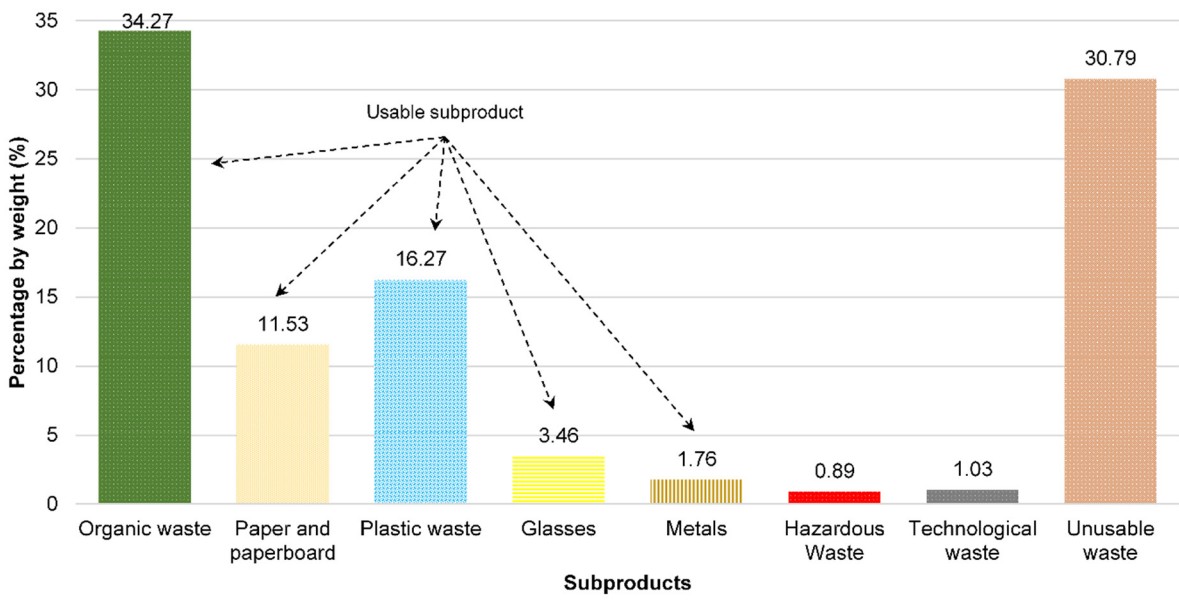

**Figure 9.** Composition of MSW deposited in the FDS of Reforma, Chiapas.

### 3.2. Analysis of Distances and Restrictions

The FDS of Reforma, Chiapas, was located without considering the Mexican environmental regulations, as well as technical criteria for location. In the regional context, the FDS is partially compliant because it is located at long distances from airports, areas of natural importance, and geological fault lines. For example, the closest airports are located more than 40 km to the northeast and southeast, on land belonging to other municipalities or states of the Mexican Republic. Areas of natural importance, such as marshes, mangroves, estuaries, swamps, wetlands, and others, are located northeast of the study area, more than 30 km in the contiguous state of Tabasco, Mexico. These great distances do not represent a risk to environmental elements (Figure 10a).

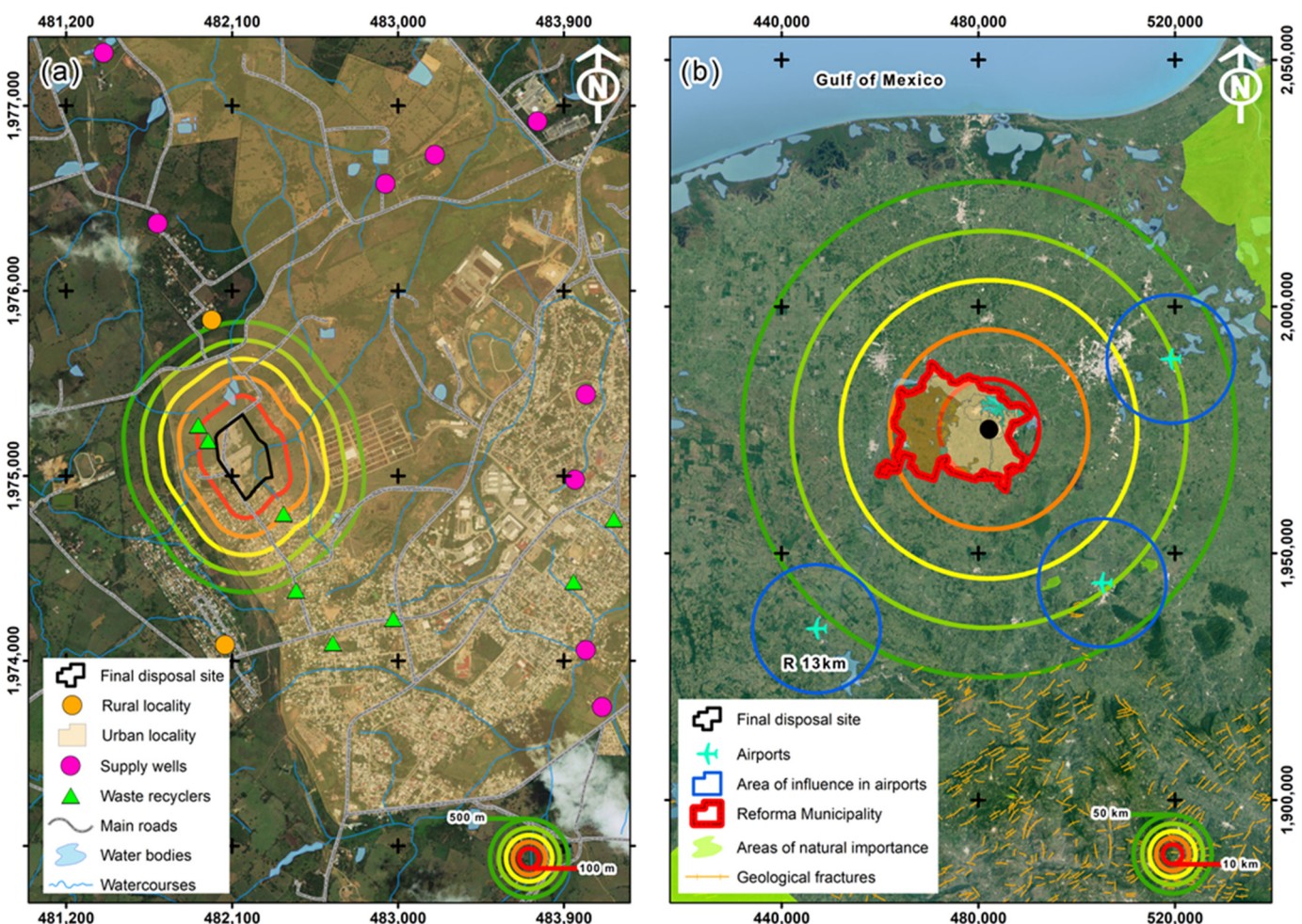

**Figure 10.** Spatial analysis of distances and restrictions: (**a**) local scale; (**b**) regional scale.

In the local context, the breaches are notorious, particularly the proximity to urban and rural settlements, since the distances are less than 500 m. This causes disgust among the nearby residents due to the generation of bad odors, dust, and infestations of rodents or insects. Additionally, due to the constant rains in the region, as well as groundwater tables close to the surface, the existence of ephemeral streams and bodies of water is common, which produce flood zones that affect the stability of the works carried out in the FDS (Figure 10b). The presence of these aspects can disturb the proper operation of the FDS, so it must be addressed through compensatory or mitigation measures, such as those indicated in previous paragraphs (natural barriers, construction of storm drains, and adaptation of roads).

Other technical restrictions in the local context are partially met. For example, the water extraction wells are located at more than 500 m, complying with Mexican regulations. The roads are accessible and close to the FDS, which allows a constant arrival of waste collection vehicles, but also causes dust and the deterioration of paved roads.

It is important to mention that many of the breaches in the local context are due to the lack of urban policies in the study area, which are reflected in the housing conditions. According to Jaramillo [60], it is common for the surroundings of the FDS to be areas where the economically poorest population lives, which increases the degree of deterioration of all sanitary conditions and also devalues the properties. This constitutes an obstacle to the urban development of cities and towns. Additionally, segregators and recyclers who are dedicated to the purchase and sale of materials obtained in the FDS also settle near the FDS, and they precariously build their homes and expand the deterioration of the neighborhood.

### 3.3. LULC Analysis

The population growth of the study area, directly or indirectly, has promoted the enlargement of the urban sprawl in the municipal seat of the study area, which has led to very important changes in the landscape and physical structure of the region. Only in the LULC analysis area, which includes 3 km around the FDS (equivalent to 452.21 Ha), urban/anthropic land use has increased at a rate of 5.82% per year, going from 43.47 Ha in 2005 to 107.52 Ha in 2021 (Figure 11a,d), but with significant increases from 2013 (Figure 11b,c). These rates are similar to those reported by Silva et al. [61] and Ramos et al. [62], for other cities larger than the studied area, such as Tuxtla Gutiérrez in Chiapas or Huimanguillo in Tabasco, both in Mexico, which have an industrial or commercial vocation. This information can be seen in Table 2.

**Table 2.** Increases and losses in land use for the period 2005–2021.

| Coverage | 2005 | | 2021 | | Increments | Losses | Persistence | Change Rate (%) |
|---|---|---|---|---|---|---|---|---|
| | Ha | % | Ha | % | | | | |
| Water | 5.00 | 1.11 | 3.29 | 0.73 | 1.88 | 3.58 | 1.41 | −2.57 |
| Urban/Anthropized | 43.47 | 9.61 | 107.52 | 23.78 | 81.45 | 17.41 | 26.06 | 5.82 |
| Tree coverage | 79.68 | 17.62 | 53.57 | 11.85 | 33.77 | 59.88 | 19.79 | −2.45 |
| Shrub and herbaceous coverage | 324.06 | 71.66 | 287.83 | 63.65 | 66.29 | 102.52 | 221.54 | −0.74 |
| Total | 452.21 | 100.00 | 452.21 | 100.00 | 183.40 | 183.40 | 268.81 | |

The use of urban/anthropized land has practically gained coverage of three other land uses in the study area. Tree cover has decreased at a rate of −2.45% in the global period of 2005–2021, being stronger for the period 2015–2021 with a negative rate of −12.40%. For the periods 2005–2013 and 2013–2015, tree cover has presented small gains of 3.12% and 7.87%, respectively. On the other hand, the herbaceous and shrubby cover has also varied over the years, with rates lower than −5.40% per year. The water class has presented variable changes in the study area, which do not depend on anthropic activities such as waste disposal, but rather by rainfall and groundwater tables near the surface. The changes have been −2.57% per year for the global period 2015–2021.

The factors that have caused these notable changes in the study area are mainly due to the increase in the number of houses, roads, industrial zones, shopping centers, and schools, which are directly influenced by population growth and also by bad urbanization policies. According to statistics from INEGI [63] and INEGI [64], the number of inhabitants in the study area went from 40,711 in 2010 to 44,829 in 2020; likewise, the number of commercial and service establishments increased by 32.37%, going from 1364 in 2010 to 2017 in 2022. Some strategies to improve the current conditions in the study area should be based on monitoring the regulation of territorial and environmental regulations and health risk studies.

### 3.4. Landfill Gas Dispersion Analysis

In the local context, particularly in the vicinity of the FDS, waste degradation processes and the dispersion of polluting emissions are notorious. Regarding landfill gas and $H_2S$, the modeling results for the 2018–2020 period indicate that the dispersion of gases from the FDS occurs in the southwest direction for both climatic seasons. The dispersal plumes can reach a range of at least 1000 m in the direction of the prevailing winds, where the concentrations that can be found range from 100 to 8725 μg/m$^3$ (0.15 to 13.33 ppm) for landfill gas, and from 0.01 to 0.35 μg/m$^3$ ($7.19 \times 10^{-6}$ to $2.52 \times 10^{-4}$ ppm) for $H_2S$ (Figure 12a,b). These concentrations are low compared to those reported in other scientific works. For example, in the works of Castillo et al. [45] and Araiza and Rojas [46], where they also used similar mathematical dispersion models, the concentrations of landfill gas and $H_2S$ found are higher than 27,000 and 1.00 μg/m$^3$, respectively. This difference in concentrations is due

to multiple factors, such as the amount of waste deposited in the FDS, organic matter degradation processes, and weather conditions, as well as waste management within the disposal cells.

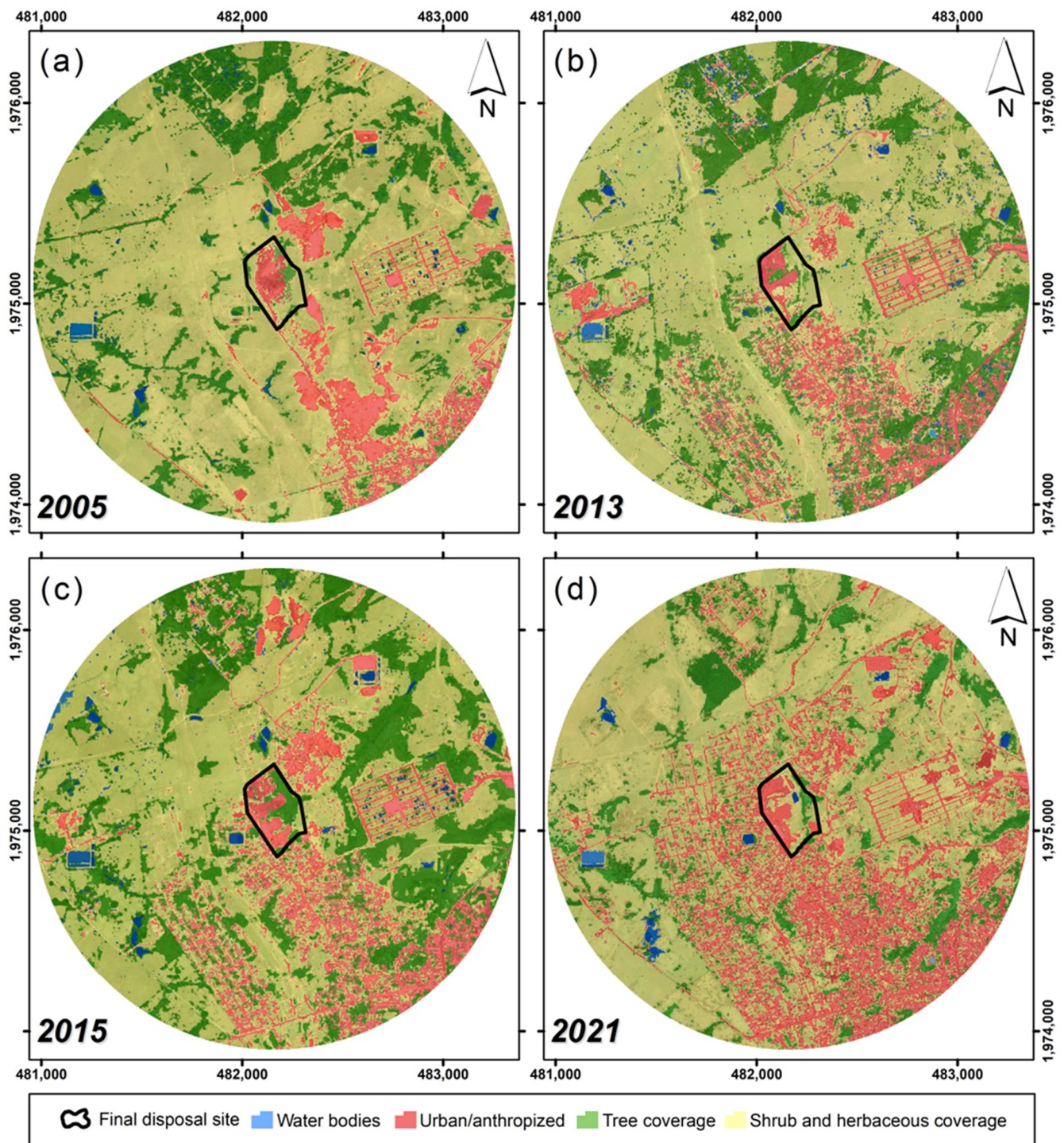

**Figure 11.** LULC analysis in the area of influence of the FDS: (**a**) year 2005; (**b**) year 2013; (**c**) year 2015; (**d**) year 2021.

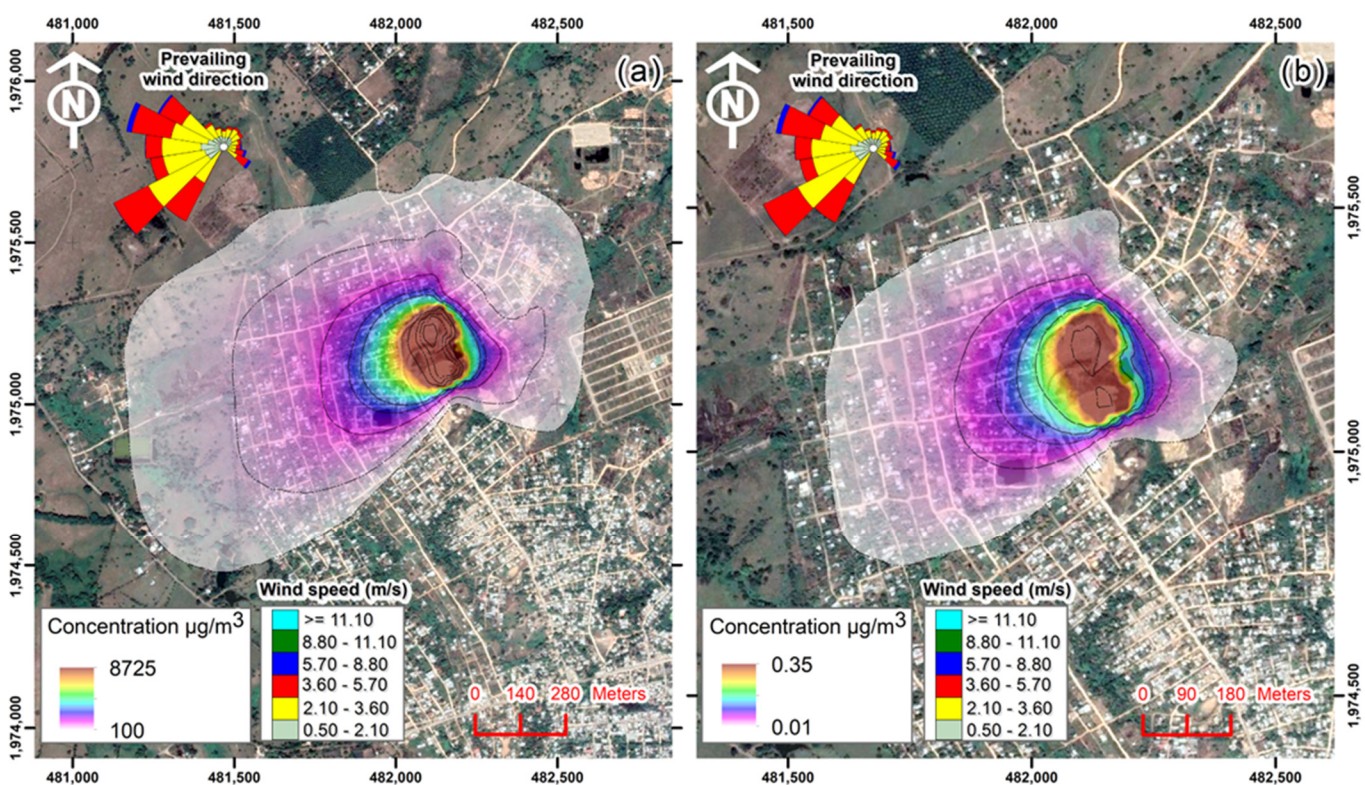

**Figure 12.** Dispersion of gaseous emissions in the FDS of the study area: (**a**) landfill gas; (**b**) $H_2S$.

Despite the low concentrations found in this work, the effects of gaseous emissions can be local and global. According to Themelis and Ulloa, [65], the effects of landfill gas, particularly the major component (methane), are due to its global warming potential when it escapes into the atmosphere, which is 23 times greater than that of $CO_2$. According to Daskalopoulos et al. [66], Köfalusi and Encarnación [67], and Tagaris et al. [68], other effects include fires and explosions within a poorly operated FDS, which occur when the major component of landfill gas (methane) is between 5% and 15% in volume, or as a result of the intentional burning of waste. Regarding $H_2S$, it is important to highlight its low olfactory threshold, which is only 0.65 $\mu g/m^3$ or less, making it detectable in places very close to the FDS, causing olfactory unpleasantness. These scenarios occurred in the study area, so it is essential to improve the operation of the FDS through good operation strategies, such as improving the frequency of waste coverage, using natural tree barriers, or even the use of burners to convert the major landfill gas (methane) emissions to $CO_2$.

*3.5. Leachates Dispersion Analysis*

The FDS leachates are produced mainly by the humidity of the waste and by the ingress of water from constant rainfall in the study area. Unfortunately, due to the lack of waste coverage and leachate control structures in the first years of operation of the FDS, leachate continues to be discharged. According to the work carried out in the field, as well as other studies in the region (see [4,34]), the deposited MSW is composed mainly of organic matter (34.27%), which causes the high organic loads of the leachates produced. A small fraction of the dumped waste is also mixed with metals (1.76%) and hazardous (0.89%) and technological waste (1.03%), which also contribute to the toxicity of this liquid.

The hypothetical modeling carried out in this work used Chloride ($Cl^-$) as the pollutant emitted by the FDS, considering the emission as an instantaneous point source. This contaminant was selected because it produces feathers of great extension, while, in addition, being a non-reactive contaminant, dilution is the only attenuation mechanism [69]. Other contaminants, such as total dissolved solids (SDT), organic matter in the form of chemical

oxygen demand (COD), or heavy metals, can be attenuated in the first layers of the soil, due to the physical–chemical processes and the biological activity in that area, which can disintegrate or retain many compounds [70].

Figure 13a shows the dispersion of leachate infiltrating into the soil. It is observed that the plume moves in the east direction, following the lines of groundwater flow in the region. Figure 13b shows the concentrations as a function of distance and modeling time. The highest concentration peak occurs in year 1 (12,270 mg/m$^3$), having a range of 50 m. The lowest concentration peak occurs in year 20 (1080 mg/m$^3$), with a hypothetical range of 450 m. Beyond these distances, the concentrations of the pollutant emitted will be lower.

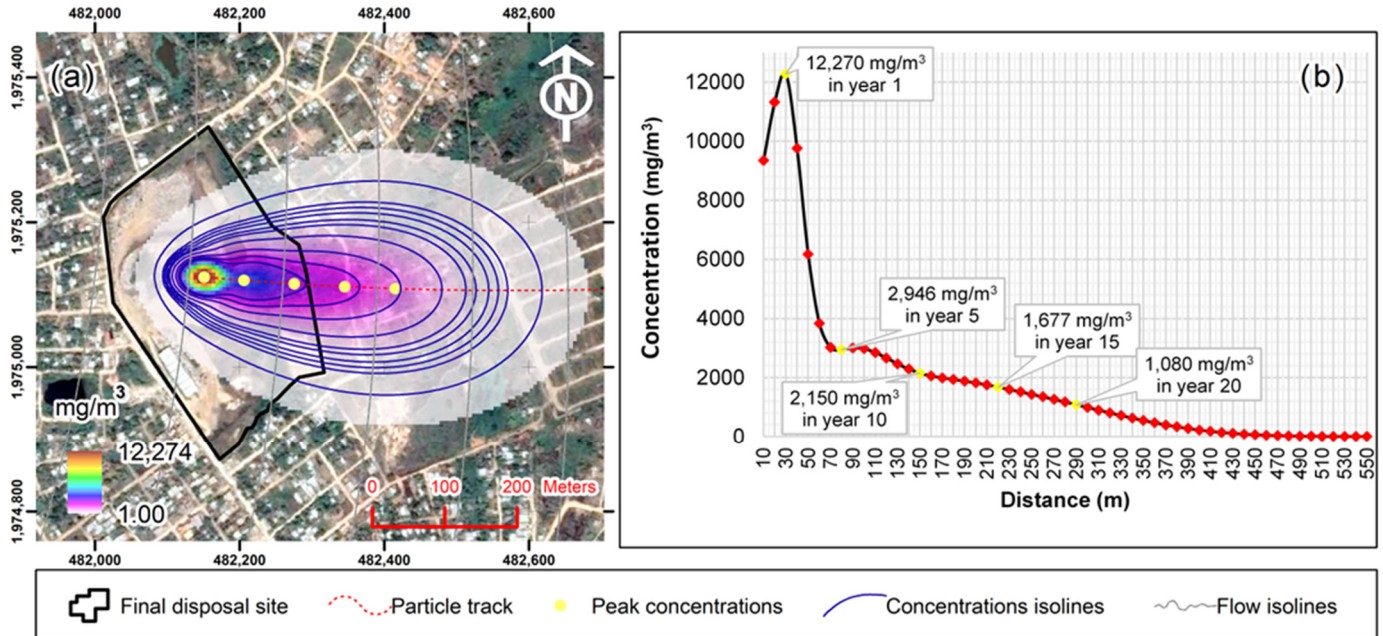

**Figure 13.** Modeling of liquid emissions in the FDS: (**a**) conservative pollutant dispersion; (**b**) behavior of concentration vs. distance.

Regardless of the concentrations presented in this work, it is important to consider the spatial scope that such emissions may have, since in the medium or long term they may affect the water reservoirs of the study area. These data are important for decision makers to analyze the feasibility of the FDS in future years.

### 3.6. LST Analysis

The determination of LST in this paper was important due to the formation of microclimatic zones 500 m around the FDS. In this work, significant temperature differences were found for the period 2005–2021, ranging from 8.37 °C for the summer season to 2.49 °C for the winter season. In Figure 14, the temperature variation between the closest and distant areas to the FDS can be clearly seen. The areas with higher temperatures always occurred within the FDS. According to Lacoboaea and Petrescu [22], this occurs due to the decomposition process of the organic matter present in the dumped waste (aerobic and anaerobic fermentation), which results in the generation of landfill gas.

Figure 14a,b show the highest temperatures found in the analyzed period (35.01 °C in summer and 27.36 °C in winter), which correspond to the most precarious operating phase of the FDS. Additionally, these high temperatures are also related to the drastic change in land use prior to the year 2005. Figure 14c–f show temperatures lower than those presented in 2005, which is due to partial compliance with Mexican regulations and the application of good operating practices, for example, the use of daily waste coverage and the restriction of burning the waste. This is also influenced by the presence of vegetation surrounding the

FDS, since, according to Pokorny et al. [71], trees offer solutions for soil cooling and local climate regulation due to their ability to capture and redistribute the sun's energy.

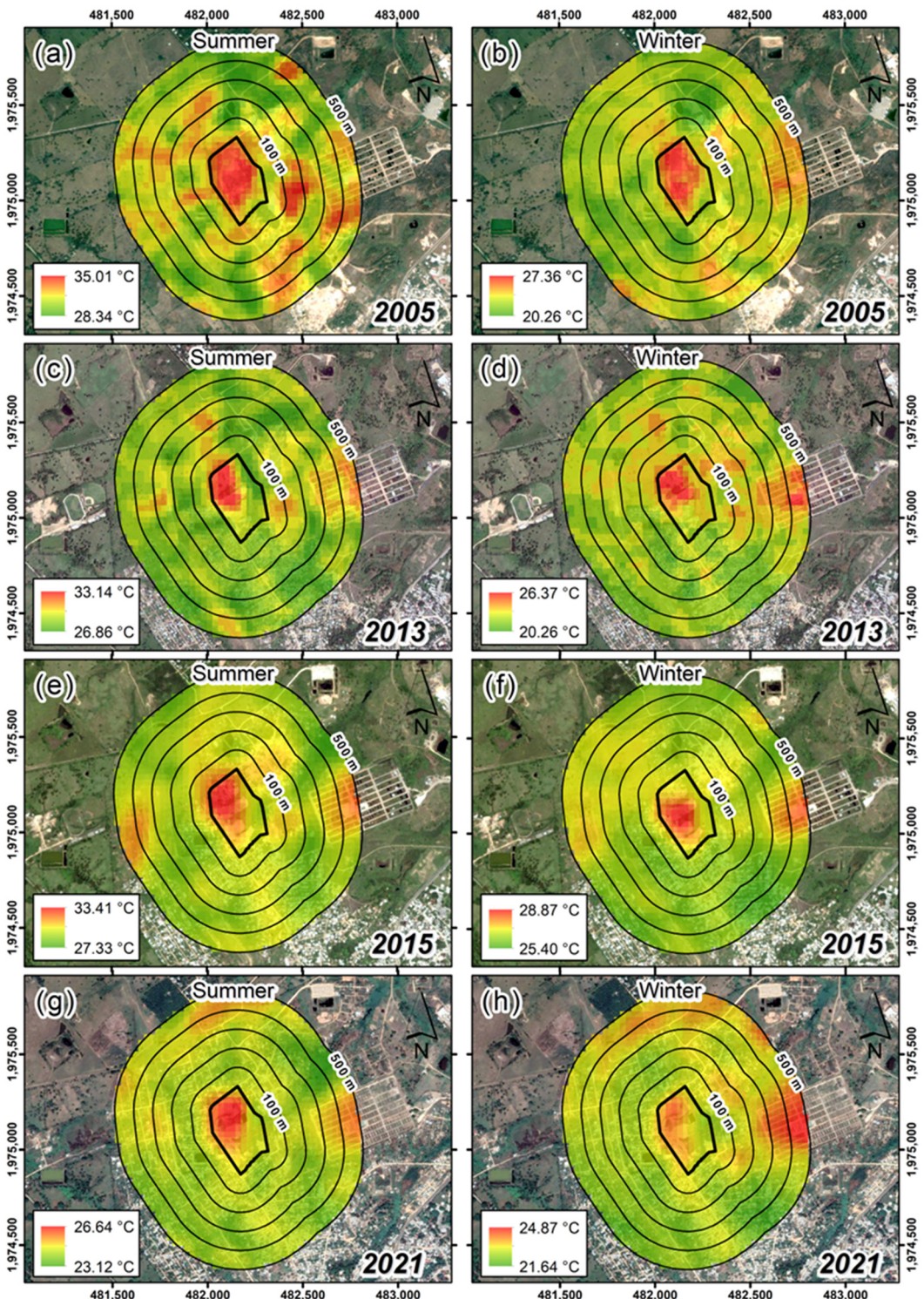

**Figure 14.** LST modeling in the surroundings of the FDS for the period 2005–2021: (**a**,**c**,**e**,**g**) temperatures in summer season; (**b**,**d**,**f**,**h**) temperatures in winter season.

Finally, in Figure 14g,h, more recent information on the temperature behavior in the surroundings of the FDS is presented. There is a difference of 1.77 °C for the highest temperatures (26.64 °C in summer and 24.87 °C in winter). In addition, the radiation source

of the highest temperatures no longer occurs within the FDS but is rather due to human settlements and their anthropic activities.

It is important to indicate that the temperature maps shown in this work will be useful for decision makers, since they can not only be used to mark the limit of the zone of thermal influence around the MSW dump sites but also to study its seasonal fluctuations that include land uses other than that of the FDS. In addition, these data can be incorporated into other studies, such as the selection of MSW management infrastructure.

### 3.7. Recommendations in Decision Making

This paper presents different spatial analysis techniques that allow visualization of the behavior of uncontrolled emissions from the FDS and regulatory non-compliance. The results presented can be used for decision making. For example, with respect to the location and current operation of the FDS, non-compliance with Mexican regulations is not possible, for which reason negotiations can be started for closure or relocation. Regarding leachate emissions, studies must be conducted promptly in water supply wells close to the FDS, to correlate emissions with environmental and human health effects. Regarding gaseous emissions, actions that can reduce environmental impacts include the placement of live barriers around the FDS, coverage of waste dumped more frequently, and placement of landfill gas burners. In terms of land use planning, it is essential for decision makers to manage studies to establish a new FDS, as well as to incorporate these priority issues in territorial risk management processes.

Finally, it is also important for decision makers in the government sector to approach the academic, social, and industry sectors to agree on municipal strategies, such as developing local regulations and designing and acquiring infrastructure for waste collection, among others.

## 4. Conclusions

In this study, the operation of an MSW FDS in Chiapas, Mexico, was evaluated using several spatial analysis techniques and GIT tools. Although in this work only the anthropogenic actions that occur in the study area are empirically modeled, the results provide useful information for decision making.

The analysis of distances and restrictions showed that the FDS partially followed Mexican regulations. In the regional context, the FDS complies because it is far-found environmental structures which may be affected, such as water extraction wells, rivers, or areas of natural importance. In the local context, the FDS does not comply because it is used incorrectly. In addition, several important works still need to be incorporated, such as storm drains, adaptation of roads, cells for emergencies, and natural barriers. In the event that these control structures are not incorporated in the medium term, scenarios of physical instability in the waste dumping processes, and severe affectations in surrounding populations, may arise.

The GIT tools and techniques used in this paper allowed visualizing the movement and spatial scope of the most important sub-products generated by MSW, such as leachate and landfill gas. Other changes that occur in the regional context, which are difficult to identify with the naked eye, such as LULC or LST, were also detected. Additionally, the use of these tools entailed advantages such as simple procedures, use of free or easily accessible data, and the possibility of replicating in other places.

**Author Contributions:** Conceptualization, Juan Antonio Araiza-Aguilar; methodology, Juan Antonio Araiza-Aguilar and Hugo Alejandro Nájera-Aguilar; investigation, Rubén Fernando Gutiérrez-Hernández; writing—original draft preparation, Carlos Manuel García-Lara; writing—review and editing, María Neftalí Rojas-Valencia. All authors have read and agreed to the published version of the manuscript.

**Funding:** This research received no external funding.

**Institutional Review Board Statement:** Not applicable.

**Informed Consent Statement:** Not applicable.

**Data Availability Statement:** Not applicable.

**Acknowledgments:** The support granted by the Institute of Science, Technology and Innovation of the State of Chiapas, for the preparation of this paper, is appreciated. We also thank the National Water Commission, as well as the professors from the University of Science and Arts of Chiapas, Juan Luis Escobar Hernández and Saul López Aguilar, for providing data to prepare the spatial analysis of this work.

**Conflicts of Interest:** The authors declare no conflict of interest.

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
