# Peer review of "Analysis of a Municipal Solid Waste Disposal Site: Use of Geographic Information Technology Tools for Decision Making"

_ijgi, doi:10.3390/ijgi12070280_

Round 1

Reviewer 2 Report

1. population settlements 500m Surface water 500 m Water extraction wells 500 m distance is very minimum to the final disposal site (FDS). did you investigate the health impact of water and waste pollution on the population?

2. have you taken a water sample to test the water contamination caused by  FDS. 

3. on line 320 you said they are receiving more organic waste. I am interested to know how they are dealing with organic waste. 

minor English correction is needed 

Reviewer 3 Report

This paper describes the analyses of several indicators in relation to landfills. The paper is well structured and written. 

TIG is used in the paper title and the content but the full words of this abbreviation is not given in the paper. the full words of GIT is proved in Section 2.3 but GIT is not used in the paper. Are GIT and TIG the same?

"for Decision Making" appears in the paper title. At the end of Section, the author also state that "The results of this manuscript can serve for decision-making in environmental matters". However, the paper provides nothing on " Decision Making" in the paper. The authors are suggested to include one section on details how " Decision Making" would be conducted from the results of analyses.

Round 2

Reviewer 3 Report

Section “3. Results and Discussion” contains no discussion in fact. Please change discussion to recommendations or remove discussion.